# Incorporating physics to overcome data scarcity in predictive modeling of protein function: A case study of BK channels

Erik Nordquist[1‡], Guohui Zhang[2‡], Shrishti Barethiya[1], Nathan Ji[3], Kelli M. White[2], Lu Han[2], Zhiguang Jia[1], Jingyi Shi[2], Jianmin Cui[2], Jianhan Chen[1]*

1 Department of Chemistry, University of Massachusetts Amherst, Amherst, Massachusetts, United States of America, 2 Department of Biomedical Engineering, Center for the Investigation of Membrane Excitability Disorders, Cardiac Bioelectricity and Arrhythmia Center, Washington University in St. Louis, St. Louis, Missouri, United States of America, 3 Department of Biology, Boston College, Chestnut Hill, Massachusetts, United States of America

‡ These authors share first authorship on this work.
* jianhanc@umass.edu

**Data Availability Statement:** All training data, predictions and associated scripts are available through GitHub at https://github.com/enordquist/bkpred.

## Abstract

Machine learning has played transformative roles in numerous chemical and biophysical problems such as protein folding where large amount of data exists. Nonetheless, many important problems remain challenging for data-driven machine learning approaches due to the limitation of data scarcity. One approach to overcome data scarcity is to incorporate physical principles such as through molecular modeling and simulation. Here, we focus on the big potassium (BK) channels that play important roles in cardiovascular and neural systems. Many mutants of BK channel are associated with various neurological and cardiovascular diseases, but the molecular effects are unknown. The voltage gating properties of BK channels have been characterized for 473 site-specific mutations experimentally over the last three decades; yet, these functional data by themselves remain far too sparse to derive a predictive model of BK channel voltage gating. Using physics-based modeling, we quantify the energetic effects of all single mutations on both open and closed states of the channel. Together with dynamic properties derived from atomistic simulations, these physical descriptors allow the training of random forest models that could reproduce unseen experimentally measured shifts in gating voltage, $\Delta V_{1/2}$, with a RMSE ~ 32 mV and correlation coefficient of R ~ 0.7. Importantly, the model appears capable of uncovering nontrivial physical principles underlying the gating of the channel, including a central role of hydrophobic gating. The model was further evaluated using four novel mutations of L235 and V236 on the S5 helix, mutations of which are predicted to have opposing effects on $V_{1/2}$ and suggest a key role of S5 in mediating voltage sensor-pore coupling. The measured $\Delta V_{1/2}$ agree quantitatively with prediction for all four mutations, with a high correlation of R = 0.92 and RMSE = 18 mV. Therefore, the model can capture nontrivial voltage gating properties in regions where few mutations are known. The success of predictive modeling of BK voltage gating demonstrates the potential of combining physics and statistical learning for overcoming data scarcity in nontrivial protein function prediction.

**Funding:** This work is supported by National Institutes of Health through grant R35 GM144045 (to J. Chen) and RO1 NS060706, R01 HL070393, and R01 HL142301 (to J. Cui). EN was also supported by National Research Service Awards T32 GM008515 and T32 GM139789 from the National Institutes of Health. The funders had no role in study design, data collection and analysis, decision to publish, or preparation of the manuscript.

**Competing interests:** The authors have declared that no competing interests exist.

## Author summary

Deep machine learning has brought many exciting breakthroughs in chemistry, physics and biology. These models require large amount of training data and struggle when the data is scarce. The latter is true for predictive modeling of the function of complex proteins such as ion channels, where only hundreds of mutational data may be available. Using the big potassium (BK) channel as a biologically important model system, we demonstrate that a reliable predictive model of its voltage gating property could be derived from only 473 mutational data by incorporating physics-derived features, which include dynamic properties from molecular dynamics simulations and energetic quantities from Rosetta mutation calculations. We show that the final random forest model captures key trends and hotspots in mutational effects of BK voltage gating, such as the important role of pore hydrophobicity. A particularly curious prediction is that mutations of two adjacent residues on the S5 helix would always have opposite effects on the gating voltage, which was confirmed by experimental characterization of four novel mutations. The current work demonstrates the importance and effectiveness of incorporating physics in predictive modeling of protein function with scarce data.

## Introduction

Generating quantitative models linking the sequence of a protein to its function remains a grand challenge in computational biophysics. Machine learning has enabled enormous advances in many key fields [1–3], most notably for predicting folded protein structures from their sequences [4–8]. There is significantly less data available describing how mutations affect the function of a specific protein, especially for complex systems like transmembrane (TM) ion channels [9–11]. Here, high-throughput approaches for functional characterization is not available and functional characterization of mutant proteins is laborious. For example, the available number of mutants with functional data is usually limited to a few hundred even for some of the most important ion channels [12–14]. Furthermore, these mutations are often distributed very nonuniformly on the protein, concentrating on certain regions perceived to be functionally important and only involving limited types of mutations (e.g., A or Q scanning). The severe data scarcity makes it generally unfeasible to derive predictive functional models of these complex proteins using the traditional data-centric machine learning approaches [15–17]. There is a great need for developing new approaches that allow the construction of reliable predictive models of protein function using limited but precious available mutagenesis data. Such models are crucial for one to better reconcile nontrivial and convoluted effects of mutations and uncover deeper mechanistic insights into the function of the target protein.

Overcoming the data scarcity problem requires additional information from independent sources. This could include multisequence alignment and structural data [18–23]. At a more fundamental level, the correlation between protein sequence and function is determined by the laws of physics, albeit the correlation is complex and of extremely high-dimension. In principle, physics-based molecular dynamics (MD) simulations could be used to generate complete trajectories and derive any kinetic and thermodynamic properties required for predicting the functional effects of any mutation, given a realistic energy function and unlimited computational power [24–28]. Indeed, physics-based MD simulations have been a workhorse for computational studies of protein function, and their reach has expanded greatly in recent years, thanks to advances in efficient GPU-accelerated algorithms [29–33], special-purpose

Anton supercomputers [34–36], and improved general-purpose protein force fields [37–39]. Nonetheless, limitations on the force field accuracy and computational cost persist, and these limitations prevent direct application of physics-based modeling and simulation for general prediction of protein function. Instead, deriving predictive models of protein function will require integration of (sparse) functional data, sequence, structure, and physics-based properties.

One simple and attractive approach of incorporating physics in machine learning to first use physics-based modeling and simulation to annotate the effects of each mutation on the protein structure and dynamics as well as conformational energetics. These physics-derived features could then be supplied to statistical learning algorithms to uncover transferable correlations for functional prediction even with sparse mutational data. Machine learning methods have been used to combine the general semi-empirical and physics-based terms with thermo-stability data of proteins to build more powerful models of thermostability [40–44]. In a recent review of protein stability prediction, it was observed that the general performance of a wide range of models has stagnated at root-mean squared error (RMSE) $\approx$ 1 kcal/mol and R $\approx$ 0.4–0.6 [45]. The integration of physics-derived features using machine learning has been extended to a wide array of protein-specific targets where sufficient data exists, such as in GPCRs [46,47], amyloid formation [48–53], and binding affinity in several proteins such as Hsp70 chaperones [54,55], SH3 domains [56], or major histocompatibility complex [57]. Incorporating MD-derived data in machine learning to predict function is an increasingly popular approach [58–61]. However, quantitative prediction of protein-specific functional properties, particularly large proteins involving multiple conformations, such as KCNQ1 channels [62], remains challenging.

In this work, we focus on the big potassium (BK) channel, also known as MaxiK, *KCNMA1* or *Slo1*, which plays a central role in regulating $K^+$ influx during membrane repolarization and is expressed in a variety of tissues including in smooth muscle cells, cardiac muscles, skeletal muscles, and neurons [64–67]. BK channel mutants have been implicated in disease, including stroke, hearing loss, asthma, and a wide array of neurological disorders [67–73]. A reliable prediction of the functional effects of mutation would allow electrophysiologists to better target their efforts in characterizing novel mutations, and aid in the diagnosis of the molecular mechanism of novel mutations as they are identified [68,69]. Functional BK channels are homo-tetramers, with each monomer composed of three major domains: voltage-sensor domain (VSD), pore-gating domain (PGD) and cytosolic tail domain (CTD) (Fig 1A). On the N-terminal side, the VSD consists of the four canonical TM helices S1-S4 plus an additional S0 helix. The PGD contains TM S5 and S6 and the selectivity filter. Finally, the CTD contains two RCK domains (RCK1 and RCK2) per monomer and is responsible for sensing intracellular $Ca^{2+}$ binding. BK channels can be activated independently by $Ca^{2+}$ and membrane voltage [74–76]. Cryo-EM structures have been determined for BK channels in both $Ca^{2+}$-bound (open) and -free (closed) states [63,77–79]. In contrast to Kv channels, the VSDs are not domain-swapped with respect to the PGD in BK channels. Curiously, the pore of BK channels remains physically open even in the closed state, in contrast to the homologous MthK [80]. Atomistic MD simulations revealed that the deactivated pore readily undergo hydrophobic dewetting transition [81,82], forming a vapor barrier for ion permeation. A physically open pore in deactivated BK channels is consistent with several key experimental findings, particularly puzzling accessibility to pore blockers [83,84] and methanethiosulfonate reagents [85]. Extensive efforts have been dedicated to understand the function and regulation of BK channels over the last three decades [86–109]. A key experimental measure of the effect of each mutant on function is shift in the voltage required for achieving half of the maximum ionic conductance ($\Delta V_{1/2}$), as illustrated in Fig 1B for WT and V236W mutant BK channels. Our experimental study and

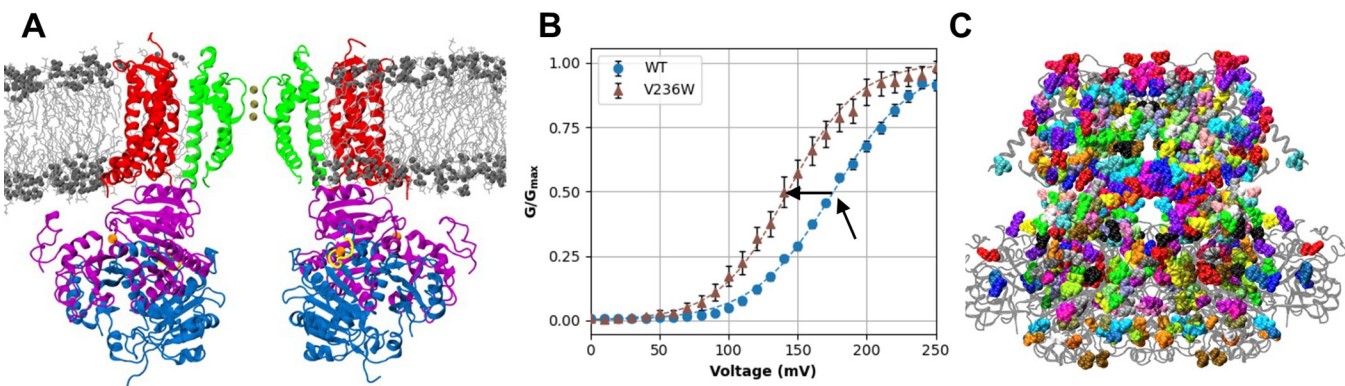

**Fig 1. Overview of the BK channel structure, voltage gating, and mutations. A)** Cryo-EM structure of the $Ca^{2+}$-bound structure of human BK channels (PDB: 6V38 [63]) embedded in a lipid bilayer. The protein is drawn in cartoon style, with the PGD colored in green, VSD in red, RCK1 in purple and RCK2 in blue. The $Ca^{2+}$-binding sites are shown in yellow with bound $Ca^{2+}$ ions shown as orange spheres. Bound $K^+$ ions in the selectivity filter are shown in gold spheres. The lipid aliphatic chains are drawn in gray bonds, with the polar head groups in dark grey spheres. This snapshot was taken from an MD-equilibrated simulation. **B)** Normalized ionic conductance-voltage (G-V) curves measured for the WT and V236W mutant BK channels. Dashed lines plot the Boltzmann fits for each curve (see Methods). The black arrows mark the WT $V_{1/2}$ as well as the shift ($\Delta V_{1/2}$) for V236W with respect to the WT. **C)** All residues with a mutation in the dataset (see Methods) drawn in different-colored van der Waals spheres. The rest of the BK channel is drawn in transparent black Cartoon.

literature search yielded 473 total mutations characterized by patch-clamp electrophysiology experiments at 0 μM $Ca^{2+}$ (see Methods), which are concentrated in the TM domains and the interface between TM and CTD (Fig 1C). Notwithstanding these extensive efforts, the molecular basis for voltage-gating and allosteric mechanisms of BK channels remain poorly understood [64,110] and no model exists that can reconcile the available mutational data and predict the functional effects of novel mutations. Note that 473 existing mutations, while an enormous body of work, only represents ~3% of single mutations to structurally resolved regions of the channel and <2% of all possible single mutations to the gene. A reliable predictive model of the voltage-gating property of BK channels is highly nontrivial due to the data scarcity problem.

To evaluate if incorporating physics could help overcome data scarcity in modeling BK voltage gating, we derive a large array of physics-based features to quantify the effects of each mutation on the structure, interaction and energetics of the channel in both open and closed states. These features were combined with additional ones that capture structure-derived properties such as secondary structure or functional domains, the change in hydrophobicity of the amino acid, and evolutionary sequence conservation. We then evaluated various statistical learning models, and the random forest model was selected as giving the best and most consistent results. The final predictive model was able to reproduce unseen experimentally measured shifts in the BK gating voltage, $\Delta V_{1/2}$, with a RMSE ~ 32 mV and correlation coefficient of R ~ 0.7, which is superior to control models trained without physics-derived features. Importantly, predictions from the final model captures some of the known but nontrivial mechanistic properties of BK channels. For example, the model captures the important role of inner pore hydrophobicity for BK activation, which is not directly encoded in any of the input features but consistent with the hydrophobic gating mechanism [81,82,111]. We further validated a prominent but curious pattern revealed by the model, that mutations of residues L235 and V236 on S5, would have opposing effects on $\Delta V_{1/2}$. We expressed four novel mutations and the measured $\Delta V_{1/2}$ agree with the prediction with a high correlation of R = 0.92 and RMSE = 18 mV. The success of a predictive model of BK voltage gating not only provides a useful tool for identifying mutational hotspots on the BK channel relevant to voltage gating, but also supports the importance and efficacy of incorporating physics to overcome data scarcity in predictive modeling of protein function.

## Results and discussion

### Physics-based features describe key properties of BK channels

We used molecular modeling and simulation to derive a large array of physics-based features to quantify the effects of all possible mutations on the structure and interactions of BK channels in both open and closed states (see Methods). These descriptors are summarized in Table A in S1 Text. A major class of these descriptors are energetics calculated using Rosetta [112–114], which quantify the relative free energy shift of the channel open-close equilibrium caused by a mutation (ΔΔG) (Fig 2A).

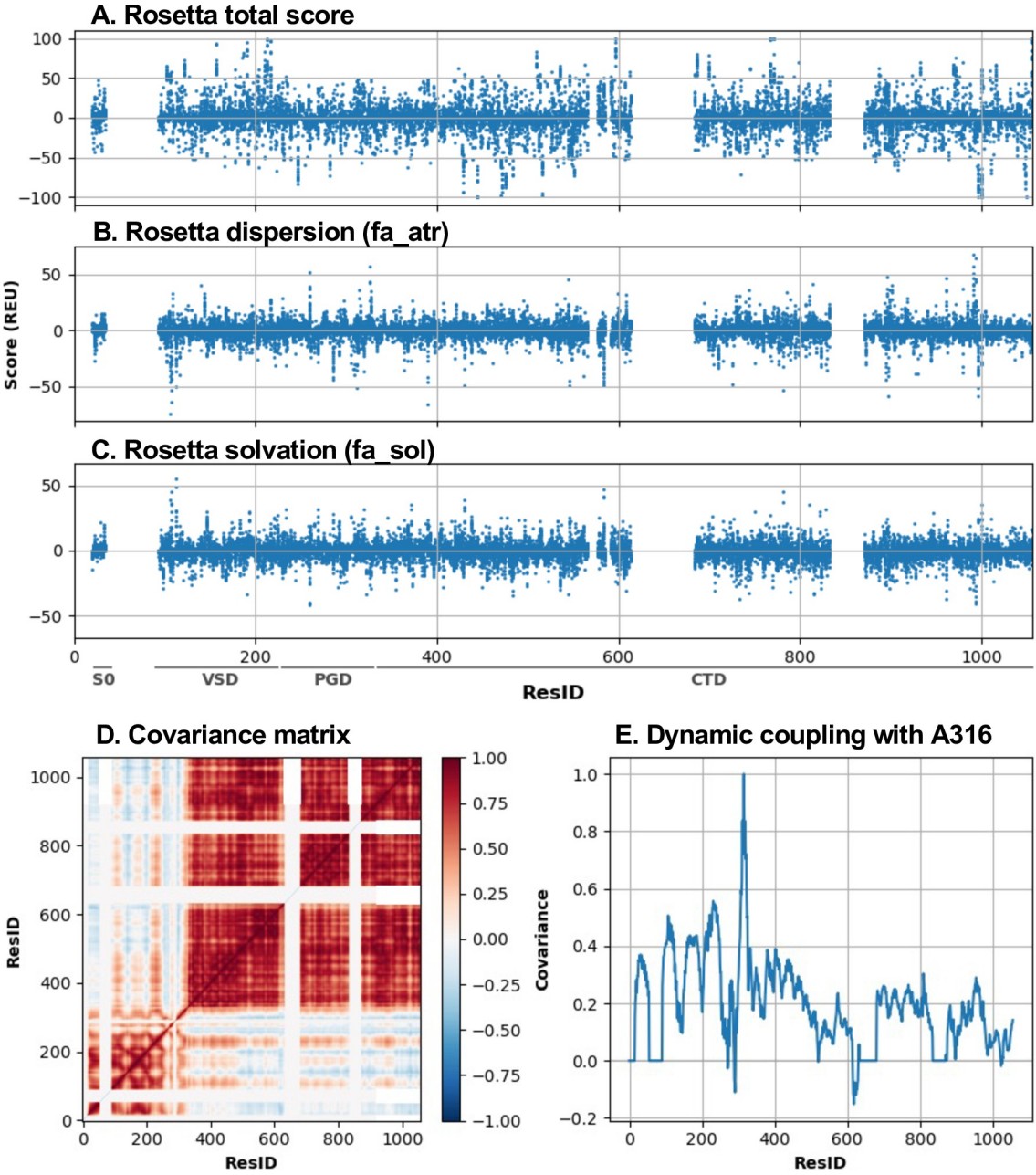

**Fig 2. Overview of key physics-based descriptors. A)** Total Rosetta ΔΔΔG scores as a function of residue number (ResID). **B)** Rosetta dispersion (fa_atr) ΔΔΔG. **C)** Rosetta solvation (fa_sol) ΔΔΔG. **D)** $C_\alpha$-$C_\alpha$ covariance matrix within the monomer in the closed state, averaged across 4 monomers, derived from atomistic MD simulation in explicit solvent and membrane. **E)** Row of covariance matrix in **(D)** corresponding to the pore-lining residue A316.

Specifically, we used the Franklin2019 Rosetta energy function [114] that include a physics-based implicit membrane model IMM1 [115]. This modified Rosetta energy function has been shown to accurately predict folding free energies of membrane protein mutants [116–119]. From the Rosetta ΔΔG calculations, we also obtained individual energy terms due to the pair-wise additive nature of the energy function. These individual terms quantify the contributions of different physical interactions and thus provide useful additional information for training the predictive model. For example, the attractive and repulsive Lennard-Jones potential terms (fa_atr, Fig 2B and fa_rep) capture favorable packing or unfavorable clashes of the mutation. The solvation free energy term (fa_sol) captures the effects of solvent exposure or burial of different side chains on the channel open-closed equilibrium (Fig 2C).

In addition, atomistic MD simulations were performed to characterize the flexibility (root-mean square fluctuation, RMSF in Table B and Fig A in S1 Text), the percent of solvent-accessible surface area (SASA; Fig A in S1 Text), and calculate covariance matrix of dynamic fluctuation (Fig 2D). The covariance matrix captures allosteric dynamic coupling between key residues, such as A316 and V278, with the rest of the channel. Here, A316 locates in the middle of the inner pore that undergoes hydrophobic dewetting transition during deactivation (Fig 2E). V278 packs against the S5 helix as part of the so-called pore helix between the filter and the S5 helix. This residue was selected because it contained several large-shift mutations and had a covariance row not colinear with the row corresponding to A316. We hypothesized that the coupled motion with the pore in the closed state would contain information relevant to predicting the gating transition. Note that MD simulations were performed for the wild-type (WT) channel in the closed state, as the dynamic properties are largely dictated by the topology and usually insensitive to single mutations. We also calculated a set of properties from the equilibrated closed state conformation to annotate the structural context of each residue site, including solvent and lipid exposure, pore-lining and secondary structure. We included the change in hydrophobicity from WT to mutant [120]. Sequence conservation was characterized using the SIFT tool [121], which incorporates genomic-level information to describe the functional impact of a mutation.

The above features describe as much as possible potentially physical consequences of mutations that may be relevant for predicting of their impacts on the open-closed equilibrium of BK channels (and thus voltage gating properties). Note that the correlation of these raw features with $\Delta V_{1/2}$ is complex and any of these features alone is insufficient to predict $\Delta V_{1/2}$ directly. For example, the total Rosetta ΔΔΔG score or its components have no direct predictive power for $\Delta V_{1/2}$, (Fig B in S1 Text). The key is thus to integrate these physical features with available experimental data using machine learning methods to uncover any hidden, non-trivial correlation between the various features and the functional output of known mutations.

## Random forest best captures key trends of available experimental data

We have measured $\Delta V_{1/2}$ of 230 single mutations, and in addition obtained $\Delta V_{1/2}$ for 243 mutations from literature (see Methods and SI; full Excel file of mutations available at https://github.com/enordquist/bkpred). These data contain mutations ranging from mild $\Delta V_{1/2} \sim 0$ mV to severe (> 50 mV) (Fig C in S1 Text (panel A)), and from all three major domains (Fig 1C), though the majority falls in the TM region and specifically the pore (Fig C in S1 Text (panel B)). Using these data from a total of 473 mutations, we trained and tested a variety of machine learning models, including linear regression with l2 regularization (Ridge regression), support vector regression (SVR), k-nearest neighbors (KNN), random forest (RF), gaussian process (GP) and multi-layer perceptron (MLP), using a grid search of the hyperparameters and 5-fold cross validation (Table B in S1 Text). The performance of various models was

mainly assessed by examining Pearson's correlation coefficient (R), RMSE, and Enrichment Factor (EF) (see Methods). The results suggest that RF performs best at balance between minimizing overfitting in training and maintaining a comparable performance on the test set with average Pearson correlation coefficient of R = 0.79. MLP with a single 100-node layer had similar performance on the test data with an R = 0.77. However, it appears to suffer from substantial overfitting, with a large discrepancy between training and test RMSEs (2 vs 30 mV). The KNN, SVR, and GP models appear to suffer less from overtraining than the RF model, but its overall performance on test data was somewhat worse than RF with an R ~ 0.73, 0.69, and 0.57, respectively. The rest of the models evaluated all performed far more poorly (see Table B in S1 Text). Note that the RF model still suffers from overfitting. The model, trained with the optimal hyperparameters (bold entries in Table B in S1 Text), achieves R = 0.97 and RMSE = 17 mV on the full training set (80% of all data) but only R = 0.79 and RMSE = 32 mV on the test set (the remaining 20%) (Fig D in S1 Text, lower right). This is likely due to the small dataset size as well as non-uniform distributions of the mutations on the protein. As illustrated in Fig C in S1 Text, existing mutations concentrate heavily on the TM domains and regions of CTD involved in interaction with the TM domains or $Ca^{2+}$ binding. Furthermore, most mutations have modest effects on $V_{1/2}$ with shifts no greater than 50 mV, while only 10% of the total of 473 mutations lead to large shifts of ±100 mV or greater (Fig C in S1 Text).

To obtain a more robust assessment of RF's performance across the full dataset, and more importantly, to better estimate its performance on unseen mutations, we resampled the full dataset and performed cross-validation on four additional 80/20 splits of the full dataset. For completeness, the performance of all models is given for all five splits. The results are shown in Fig 3 for the RF model and summarized in Table 1 for all models. The RF model's performance was relatively robust across the five splits with R = 0.54–0.80, RMSE = 30–35 mV, and EF = 2.2–3.3 (Table 1). The model appears to perform better, particularly with a higher R and EF, when the data coverage is similar to the coverage in the training set. For example, in split 2, the fraction of mutations with $|\Delta V_{1/2}| > 50$ mV is ~19%, substantially lower than across the whole dataset (~23%) or in the other four splits (23% - 26%) This is consistent with the model's performance being hindered by limited data and substantially different distributions in the training and test sets. The results on the five independent splits shows that the model's performance across the whole sequence space will not be as good as the best of these splits, but realistically can be expected to provide significant improvement from random selection of mutations. This is highlighted by the EF of 2–3, suggesting that mutations selected near the top of the predicted distribution, regardless of the semi-quantitative nature of the predictions, is sufficient to give 2–3 times greater enrichment of large shifts (e.g., more than ±50 mV) than by randomly drawing from the experimental dataset directly. The results of validation using random splits of the dataset demonstrate that, while a model trained this way is not expected to be quantitatively accurate across all unseen data, it can likely be used to scan for hotspots of large shifts or uncover prominent physical trends in mutational effects of channel function.

## Importance of physics-based features

We first evaluated the importance of various input features to the final RF model's performance by examining the mean decreases in Gini impurity scores. As shown in Fig E in S1 Text, depicting a representative feature importance, most of the top-ranking features are physics-based, including the dynamic coupling to the pore-lining residue A316, RMSF, and solvation and dispersion terms. To further test the importance of the physics-based features, we trained a best-effort control RF model using the same random forest hyperparameters but omitting all features derived from Rosetta or MD simulations. Instead, we supplement the

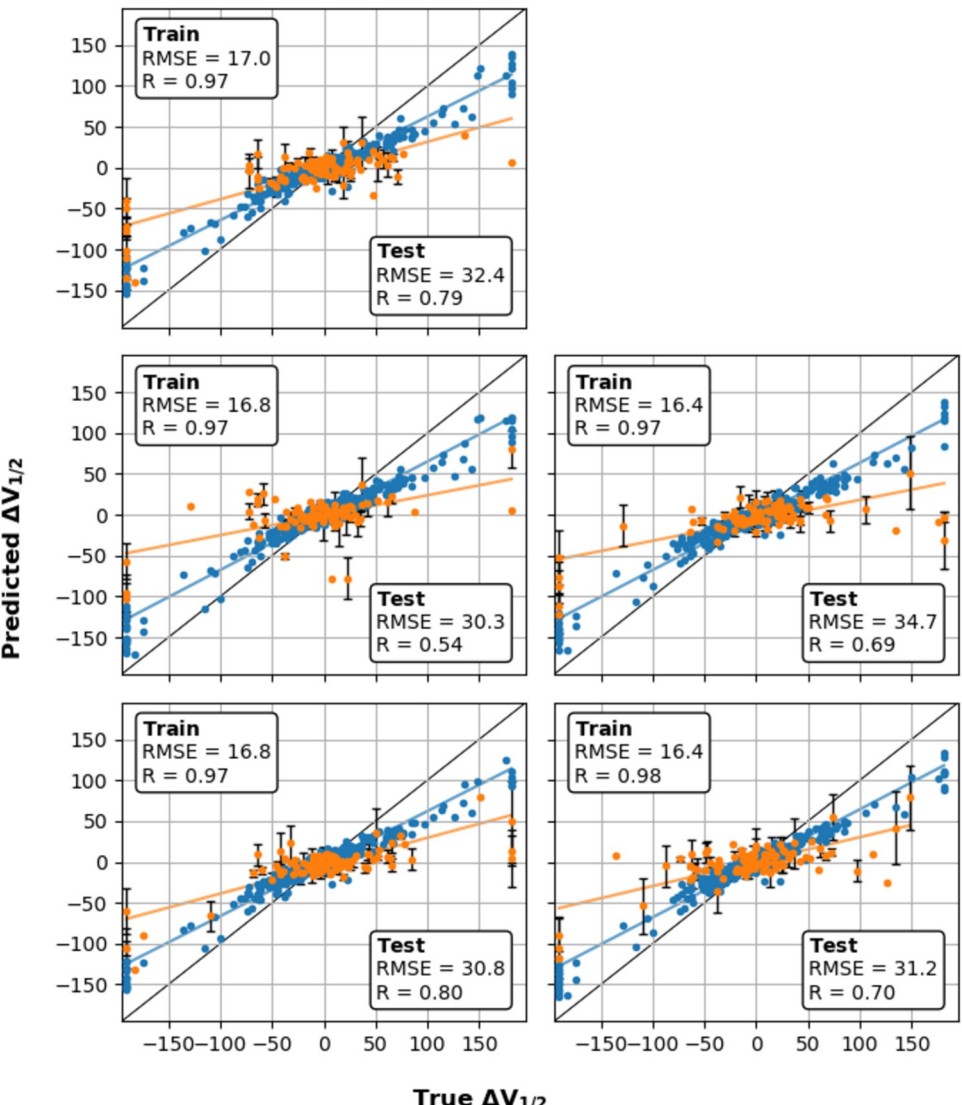

**Fig 3. Results of training and validation in 5 random train/test data splits.** Correlations of predicted and true $\Delta V_{1/2}$ for 5-fold cross-validation on 80% of dataset (blue) and independent test validation on the remaining 20% (orange). The dashed lines indicate trends for training and test, and the solid line marks x = y. The blue points show the performance on the training dataset, with overall R = 0.97–0.98, RMSE = 16–17 mV. The orange points show the independent test set with R = 0.54–0.80, RMSE = 30–35 mV.

sequence and structure features with 19 previously-published principle component descriptors [122] derived from over 500 biochemical and biophysical properties of amino acids from the AAIndex database [123]. Specifically, we use the changes from WT to mutant AAIndex values for each of these features. As summarized in Fig F in S1 Text and Table 1, the control models performed significantly worse without physics-based features, with R = 0.1–0.6, RMSE = 29–39 mV, and EF = 1.5–3.0. The average R is only 0.38, far below ~0.7 achieved with physics-based features. The models become much more sensitive to overfitting, manifested as large performance gaps on training vs test (Fig F in S1 Text). This strongly supports the importance of including physics-based features in overcoming the data scarcity problem and generating transferable predictive models of protein function.

**Table 1. Goodness-of-fit metrics of RF models trained with and without physics-based features.** Two sets of RF models were trained and validated using five independent 80/20 splits of the dataset. The models trained without MD- and Rosetta-derived features were supplemented with biochemical and biophysical features of amino acids derived from the AAIndex database [123].

| | With physics-based features | | | Without physics-based features | | |
|---|---|---|---|---|---|---|
| **Split** | **R** | **RMSE (mV)** | **EF** | **R** | **RMSE (mV)** | **EF** |
| **1** | 0.79 | 32 | 3.0 | 0.42 | 37 | 2.8 |
| **2** | 0.54 | 30 | 2.4 | 0.10 | 29 | 1.5 |
| **3** | 0.69 | 35 | 2.2 | 0.45 | 39 | 2.2 |
| **4** | 0.80 | 31 | 3.3 | 0.27 | 33 | 3.0 |
| **5** | 0.70 | 31 | 2.4 | 0.60 | 36 | 2.4 |

## Characteristics of the final RF model of BK voltage gating

Having robustly examined the RF model's performance both on fitting training data and predicting unseen data using several random splits, we trained a production RF model with the same hyperparameters but using all the available data. This was deemed prudent given the limited size of the dataset. This model was used to generate predictions throughout the rest of the paper, and its performance is expected be largely similar to those trained with various 80/20 splits as reported in Table 1. The production RF model was used to predict the $\Delta V_{1/2}$ shift of all 16264 single mutations for the 856 residues present in both $Ca^{2+}$-bound and free structures, available through GitHub at https://github.com/enordquist/bkpred) as well as part of the S1 Text. In Fig 4, we compare the maximum $\Delta V_{1/2}$ shift at each residue position derived from available experimental data and RF prediction. As expected, RF predictions generally recapitulate the experimental pattern, showing that mutations near the pore or at the TM/CTD interface tend to have large maximum effects, while those in the rest of CTD have much more limited maximum impacts on voltage gating. While this agrees with the expectation that residues at or near the interface will contribute more to gating, the lack of large shifts in the CTD regions distal to the TM domains likely also reflects a lack of experimental data in these regions (Fig C in S1 Text). Furthermore, the structure of CTD is similar between the open and closed states in most parts. As such, $\Delta\Delta\Delta G$ terms calculated from Rosetta are generally of smaller

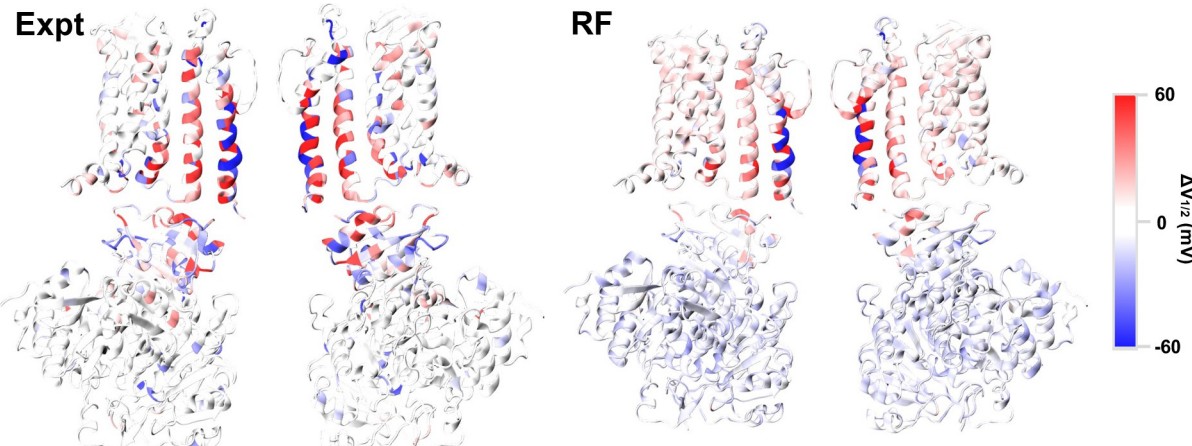

**Fig 4. Maximum experimental (Expt) and predicted (RF) $\Delta V_{1/2}$ for mutations at each position.** For each residue position, the maximum shift was selected from available experimental mutants or predicted values of all possible mutations. Only two opposing monomers are shown for clarity.

magnitude for these distal sites as compared to those in the TM region, which could presumably translate to smaller predicted $\Delta V_{1/2}$ shifts. Nonetheless, the observation that the RF model does not tend to extrapolate large shifts in distal regions of the channel is a desired behavior.

Note that, despite a respectable R of ~0.7, the final RF model tends to predict smaller $\Delta V_{1/2}$ magnitude than the true values (Fig 3). This is likely because of the imbalance of the number of mutations with small $\Delta V_{1/2}$ versus those with large $\Delta V_{1/2}$ (Fig C in S1 Text). RF models are trained to optimize the RMSE, not the slope of the correlation. Coupled with small data size, RF models could have a tendency to under predict the slope of correlation [124]. The final RF model of BK voltage gating thus appears to avoid predicting $\Delta V_{1/2}$ to be much larger than the true $\Delta V_{1/2}$. As such, the predicted $\Delta V_{1/2}$ is largely semi-quantitative and should optimally be used for determining if a mutation is likely to cause large $\Delta V_{1/2}$ shift, and if so, the direction of the shift. As will be illustrated below, even such an apparently limited predictive capability, the production RF model allows one to identify and investigate prominent features in mutational effects of BK voltage gating, providing new insights and new directions for further mechanistic studies.

The prediction error of the final RF model was further estimated using the standard bootstrap aggregation (i.e., bagging) approach [125,126]. Fig G in S1 Text compares the estimated and true errors for the test data using the same five random 80/20 splits used in Fig 3. The results show that the predicted error correlates well with the true error when the data is a part of the training set, but underestimates the true error in many cases for data outside the training set. The predicted and true errors on test set have a modest average correlation of R ~ 0.47, which is not surprising given the poor data coverage and the limited ability of the descriptors to explain all variations in the data. Despite its limitations, the error should still be useful for screening features or hotspots in the predicted distribution of mutational effects of BK voltage gating.

## RF predictions recapitulate hydrophobic gating mechanism

One of the most prominent predictions of the production RF model is that mutations in the pore region tend to have large impacts on the gating voltage. More intriguingly, the model predicts that polar and charged mutations tend to reduce $V_{1/2}$, or making the channel easier to open, while hydrophobic mutations tend increase $V_{1/2}$, or making the channel harder to open. These trends are illustrated in Fig 5 for all mutations to N, K and V in the TM domains. Note that, while N or K mutations can lead to positive $\Delta V_{1/2}$ values in the VSD or S5 (blue or red), they always reduce $V_{1/2}$ for pore-facing sites on S6 and the decreases are larger for K mutations than N mutations. In contrast, V mutations on the pore-facing sites on S6 mostly increase $V_{1/2}$ with the exception of L312V, which decreases $V_{1/2}$ as do most other mutations at 312 including L312I and L312A [92]. Importantly, while there is a relatively large number of mutations in the training data in the pore (Fig 5, left half), there is nothing specific in this data set alone that would suggest that other pore-lining residues should have similar effects on the channels open-close equilibrium. Nonetheless, these intriguing predictions from the RF model are actually consistent with the hydrophobic gating mechanism proposed for BK channels [81,82,111]. Under this mechanism, the overall hydrophobicity of the inner pore determines the ability of the pore to undergo hydrophobic dewetting and shut down ion permeation, which has been shown to correlate well with the activation voltage in a previous free energy analysis [111]. This mechanistic understanding explains why the hydrophobicity and hydrophilicity of pore mutations have large and predictable effects on BK voltage gating. It is remarkable that the RF model recovers such a physical mechanism, apparently by recognizing patterns in the physics-based features associated with known and unseen mutations in the pore.

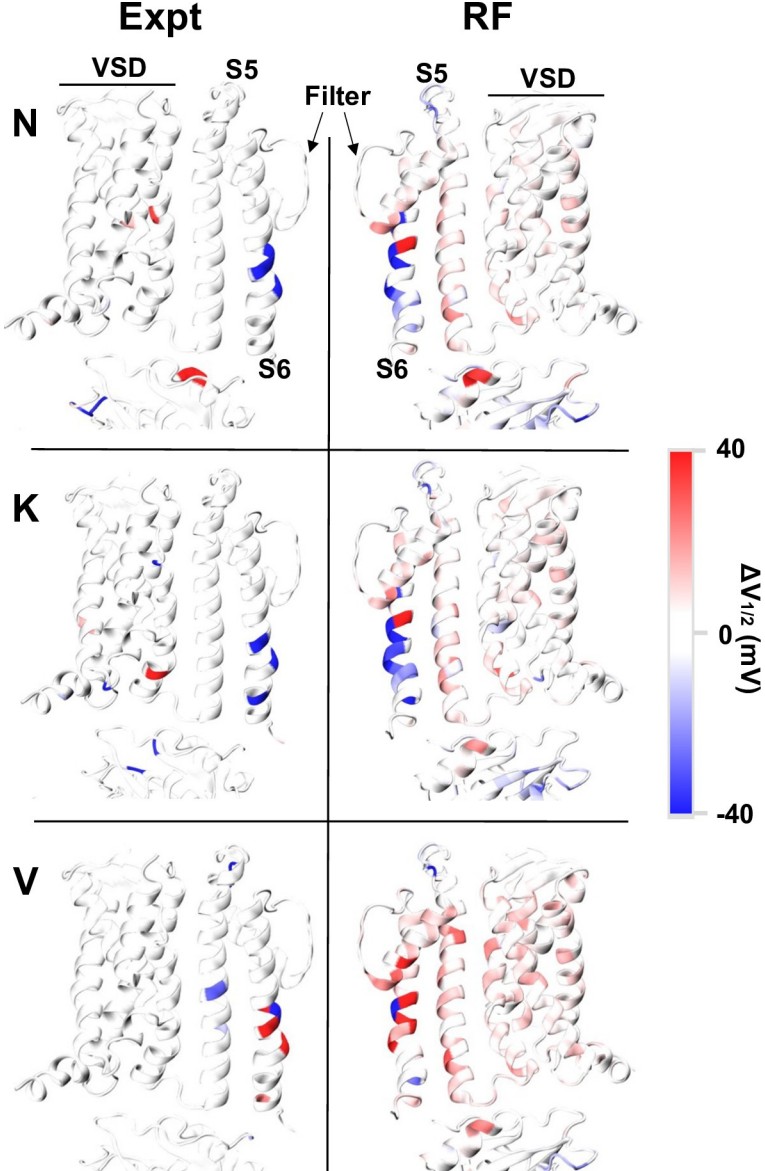

**Fig 5. Experimental (Expt) and predicted (RF) $\Delta V_{1/2}$ of N-, K-, and V-scanning mapped onto the TM structure of BK channels.** The VSD, and PGD components: S5, S6 and selectivity filter, are denoted. The two domains are facing one another as they would be in the structure (90˚ rotation), not mirror images of each other.

## RF model correctly identifies a novel trend on mutational effects of key S5 sites

Another peculiar feature predicted by the RF model involves two neighboring sites on the S5 helix (Fig 6A). As summarized in Fig 6B, all mutations of L235 are predicted to shift $V_{1/2}$ the right ($\Delta V_{1/2} > 0$), while mutation of V236 mostly shifts $\Delta V_{1/2}$ to the left ($\Delta V_{1/2} < 0$) but with smaller magnitudes. The S5 helix mediates the interaction between the VSD and the pore-lining S6 helices, and has been proposed to play a key role in the BK VSD-pore coupling [127–129]. However, the specific roles of individual S5 residues or their interactions in the coupling remain poorly understood. The experimental data set, at the time of model training, only

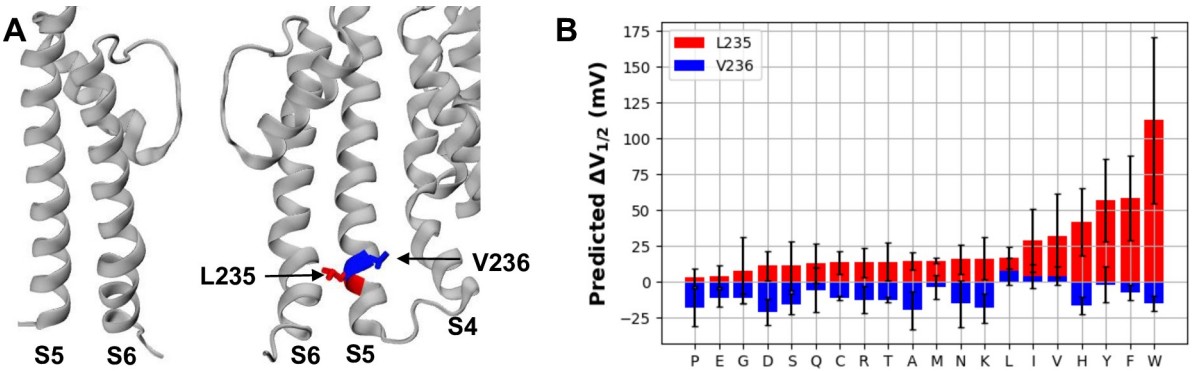

**Fig 6. S5 helix residues L235 and V236 and neighboring residues. A)** Zoomed-in view of the PGD of two monomers, with L235 and V236 labeled and colored in red and blue bonds, respectively. The PGD helices S6 and S5, as well as the contacting VSD helix S4, are labeled. **B)** Predicted $\Delta V_{1/2}$ of all mutations of L235 (red) and V236 (blue), arranged by increasing magnitude of predictions for L235X. Note that WT "mutations", L235L and V236V, reflect the inherent uncertainty of the RF model prediction.

contained three hydrophobic mutations for these sites (L235W, V236A and V236W; see Table 2). The predicted trend, if validated, thus would have profound implications on how the S5-S6 and S5-VSD interfaces control the pore-VSD coupling in BK channels. A new mutation, L235A, was published in December 2022, after the completion of our model training [127]. The experimental result confirms that replacing L with the smaller A does also increase $V_{1/2}$, but surprisingly, the shift is much larger than predicted and similar to that of the much larger W mutation.

We further expressed and measured the G-V curves of four novel mutations, namely, L235F, L235H, V236H, and V236N, with patch-clamp electrophysiology. We selected L235F because of the high predicted $\Delta V_{1/2}$ of 59 ± 29 mV and its apparent similarity to W. L235H and V236H were selected as a direct test of the predicted trend that mutations to these two adjacent sites have opposite effects on BK voltage gating. Finally, we selected V236N as representative of the effect of introducing a polar mutation with a small sidechain. The measured current traces and the fitted G-V curves are shown in Fig 7A and 7B. The experimental results confirm that both L235 mutations increase $V_{1/2}$, while both V236 mutations reduce $V_{1/2}$ (Table 2) This is fully consistent with the trend predicted by the RF model. Furthermore, the

**Table 2. Experimental and predicted $\Delta V_{1/2}$ of L235 and V236 mutations.** The four novel mutations are bolded. >150 signifies the case where the complete G-V curve could not be measured due to extremely low activity of the mutant. The WT "mutations", L235L and V236V, reflect the inherent uncertainty of the RF model prediction. "Previously acquired" denotes previously acquired data from the Cui lab that were available prior to model training. *The experimental result for L235A was published after model training, so this mutation is also a true test case.

| Mutation | Experimental $\Delta V_{1/2}$ (mV) | Predicted $\Delta V_{1/2}$ (mV) | Reference |
|---|---|---|---|
| L235L | 0 | 17 ± 8 | - |
| L235W | >150 | 113 ± 58 | previously acquired |
| L235A | >150 | 15 ± 6 | [127]* |
| **L235H** | **13 ± 6** | **42 ± 23** | this study |
| **L235F** | **56 ± 8** | **59 ± 29** | this study |
| V236V | 0 | 4 ± 6 | - |
| V236A | -38 ± 1 | -19 ± 13 | [87] |
| V236W | -43 ± 6 | -15 ± 5 | previously acquired |
| **V236H** | **-35 ± 7** | **-16 ± 6** | this study |
| **V236N** | **-11 ± 7** | **-15 ± 16** | this study |

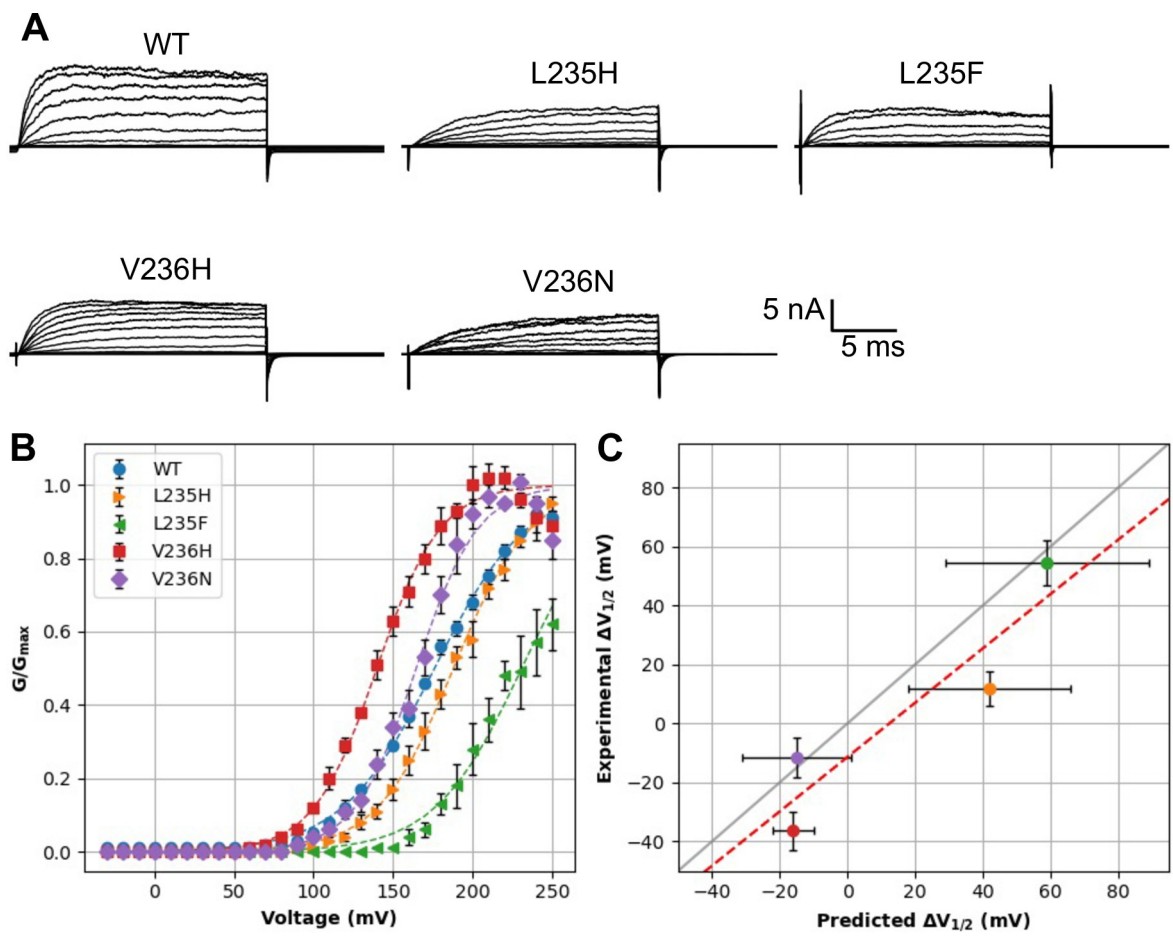

**Fig 7. Correlation of experimental and predicted $\Delta V_{1/2}$ for four novel L235 and V236 mutations. A)** Current traces for the WT and four mutant channels. **B)** Normalized conductance (G/$G_{max}$) versus voltage (V) curves for the WT and four mutants. Dashed lines denote the Boltzmann fits for each curve (see Methods). **C)** Correlation between measured and predicted $\Delta V_{1/2}$. Error bars report the predicted RF error and the propagated error from the experimental fitting, respectively. The dashed red line represents the best linear fit with R = 0.92, and the gray line plots y = x.

predicted $\Delta V_{1/2}$ values are well correlated with the experimental results, with R = 0.92 and RMSE = 18 mV (Fig 7C). The quantitative accuracy of these four novel mutations is better than expected based on cross-validation (see Fig 3 and Table 1). The strong performance here suggests that the RF model likely captures a nontrivial and robust feature of BK voltage gating. We note that only three L235 and V236 mutations exist in the experimental data set used for RF model training (Table 2). The ability of RF model to discover the distinct impacts of L235 and V236 mutations on voltage gating is noteworthy and speaks strongly to the importance and impact of including physics in deriving more reliable predictive models of protein function from small datasets.

At this point, the mechanistic basis for the opposite effects of L235 and V236 mutations on BK voltage gating is not understood. It is likely that the S5-S6 and S5-S4 interfaces have distinct roles in mediating the VSD-pore coupling (see Fig 6A). For example, because disruption of the S5-S6 packing by any mutations appears to make it harder for the pore to open (thus increase $V_{1/2}$), it is likely that the native S5-S6 interactions favor the open conformations of the pore formed by four S6 helices. Conversely, disruption of the S4-S5 packing tends decrease $V_{1/2}$, suggesting that this interface stabilizes the conductive, open-pore state. The implication is

that resting VSDs may play a role in arresting the pore in the closed state, such that weakening the link between VSD and the pore (through random L236 mutations) would generally allow the pore to open more easily. These speculations will need to be more thoroughly tested, such as using a combination of atomistic simulations and electrophysiology.

## Conclusions

Predictive modeling of how mutations may affect the function of complex proteins frequently suffers from the data scarcity problem because functional characterization of mutant proteins can be time consuming. Incorporating the physics of protein structure, dynamics and interaction could provide the additional information required for training reliable models using limited data set. In this work, we collected the voltage gating measurements of a total of 473 single mutations of the BK channel, a key target of biological and biomedical interest, and constructed a predictive model of the effects of single point mutations on BK voltage-gating, by leveraging high-resolution cryo-EM structures of both the conductive and non-conductive states and an array of physics-based features derived from MD simulations and Rosetta mutation analysis. These physics-based features annotate the dynamic and energetic consequences of all mutations, allowing the use of machine learning to better uncover patterns and correlations for predicting the voltage gating of all possible single mutation, even though the available data provides a poor coverage of the sequence with many positions seeing at most one or no mutations. The final RF model is able to predict shifts in the gating voltage of the BK channel with a correlation of R ~ 0.7 and RMSE ~ 32 mV. Replacing the physics-based features with protein-independent chemical and physical properties of amino acids through AAIndex [123] drastically reduce the accuracy of the model trained using the same functional data set.

It is noted that the final RF model remains largely semi-quantitative due to the small training set, with a tendency to under-predict the magnitude of $V_{1/2}$ shifts. Nonetheless, it is relatively reliable for identifying mutations with large $V_{1/2}$ shifts as well as the direction of the shift. As such, the model allows one to identify and investigate prominent features and hotspots in mutational effects of BK voltage gating. We illustrate that the model correctly recapitulates how changing the hydrophobicity and hydrophilicity of the pore can shift the gating voltage, consistent with the underlying hydrophobic gating mechanism of BK channels. We further investigated a curious prediction that mutations two adjacent residues on the S5 helix would always have opposing effects on BK voltage gating. Specifically, all mutations of L235 would increase $V_{1/2}$, while those of V236 would always decrease $V_{1/2}$. This prediction is confirmed by experimental characterization of four novel L235 and V236 mutations, as well as a fifth one published after the modeling training (L235A, [127]). The molecular basis of the opposing effects of L235 and V236 mutations is not yet understood, but the implication is that both S4-S5 and S5-S6 interfaces play key roles in mediating VSD-pore coupling in BK channels. It should be noted that both the hydrophobic gating and effect of L235/V236 mutations are localized to the PGD, the region with the best experimental coverage.

The final RF model may be useful for scanning for additional hotspots and provide new insights and new directions for further mechanistic studies. Therefore, we have made all the predictions of the production model available in an excel file in the SI. In addition, we compile a table of mutations implicated in *KCNMA1*-linked channelopathies [68,69,130,131], with their known functional consequences and the predicted $\Delta V_{1/2}$ (Table C in S1 Text). These mutations present significant challenges to the RF model, as the loss- or gain-of-function could result from changes in $Ca^{2+}$ or $Mg^{2+}$ sensitivity, changes in the expression level, or even to cellular effects like localization, aggregation, or degradation of the channel. These latter effects are not captured in the RF model, so we don't expect the model to be very useful beyond

scanning for hotspots in the voltage-gating pathway near the PGD and VSD interface, where many available mutations might lead to higher confidence predictions (Fig C in S1 Text). Nonetheless, the success of the modeling approach employed here demonstrates that MD simulations and point-mutation calculations in empirical and statistical potentials like Rosetta are a powerful way to annotate the effects of mutations, to allow the integration of physics training machine learning models with limited data sets. The modeling is relatively inexpensive, even for large membrane-protein systems like the BK channel, and the results can be impressively accurate for the specific purpose of identifying high-confidence trends and hotspots, even if quantitative agreement of effects to all protein sites remains elusive.

## Methods

### Electrophysiology data collection and preparation

We conducted a literature search for single mutations of the BK channel with $V_{1/2}$ recorded at any $Ca^{2+}$ concentration [86–109]. In addition, many not previously published mutations have accumulated in the Cui lab over the years. Each mutation is associated with the WT $V_{1/2}$ measurement and $\Delta V_{1/2}$. These data are included in the S1 Text as well as through GitHub (see below). Some mutations have been reported in multiple papers from different labs. For the most part, the $\Delta V_{1/2}$ values reported for the same mutant differ by less than 20 mV. In such cases, we use the data from the Cui lab whenever possible to minimize variability. In five cases, namely T189A, R201Q, R207Q, R213Q and K228Q, the differences in report $\Delta V_{1/2}$ values are larger than 20 mV and average $\Delta V_{1/2}$ values were used. In total, we collected 473 single mutations either in the hslo1 gene or compatible with the hslo1 gene that had a $\Delta V_{1/2}$ at nominally 0 μM $Ca^{2+}$ (as of September 2022). There were 201 mutations with at least one additional $\Delta V_{1/2}$ entry at higher $Ca^{2+}$ concentration, often 1–10 μM. 101 mutations had an additional measurement in the 10–100 μM $Ca^{2+}$ range. Fewer than 100 had a reading at three or more $Ca^{2+}$ concentrations simultaneously. We included mutations that had $\Delta V_{1/2}$ too large to be recorded, often because the mutant's $V_{1/2}$ was above 300 mV or below -200 mV. We used flag values set to the largest $\Delta V_{1/2}$ in the data set (370 and -300 mV, respectively) for all mutations with very low or very high conductance. Furthermore, we applied a quashing function,

$s\left(\Delta V_{1/2}\right) = \Delta V_{max} \cdot \tanh\left(\frac{\Delta V_{1/2}}{\Delta V_{max}}\right)$, with $\Delta V_{max} = 200$ mV, so that the magnitude of $\Delta V_{1/2}$ values would maximize at 200 mV. Note that the squashing function compresses the large shifts as they grow above 100 mV, while having little or no effect on shifts less than 100 mV (Fig H in S1 Text). The distribution of the final squashed $\Delta V_{1/2}$ values is shown in Fig C in S1 Text.

### Structure preparation and Rosetta energy calculations

We used PDB entries 6V3G and 6V38 as starting structures for the $Ca^{2+}$-free (closed) and -bound (open) BK channels [63]. Only segments present in both structures were included, assuming mutations on unstructured segments would have minimal energetic effects, leaving 856 residues. We used the Positioning of Proteins in flat and curved Membranes server [132] to obtain the membrane-aligned pdb-format structure, then used the Rosetta tool "mp_span_-from_pdb" to generate a Rosetta TM span file. We used the "clean_pdb.py" script included in Rosetta tools to satisfy Rosetta's internal formatting conventions. Residue IDs shown in this work follow the PDB numbering unless otherwise noted. The formatted, initial PDB structures were subjected to a relaxation with the fast relax protocol prior to the mutation energy calculations (see below), in which only sidechains were allowed to move.

We used PyRosetta4 [113] and the franklin2019 Rosetta score function for membrane proteins [114,116] to calculate the energetic effects of point mutations on BK channels. The

franklin2019 score function is identical to the standard Rosetta Energy Function 2015 [112], except for the addition of an implicit membrane solvation term, which accounts the free energy cost of transferring the residue from an aqueous to lipid environment. The latter is based on the implicit membrane model IMM1 [115]. The "predict_ddG.py" script released with PyRosetta was used for calculating the folding free energy, $\Delta G_{fold}$, for both open and closed conformations. Mutations were first relaxed using the fast relax protocol, where side-chains within 8 Å $C_\beta$-$C_\beta$ distance cutoff were allowed to repack. The raw total Rosetta energies were first smoothed before calculating the mutational effects, to prevent unphysically large energies from yielding unreliable differences. The squashing function used is:

$s(score) = E_{max} \cdot \tanh\left(\frac{score}{E_{max}}\right)$, with $E_{max}$ = 50 Rosetta Energy Units (REUs). Application of the squashing function truncates large scores arising from steric clashes to a maximum of 50 REUs, while having minimal effect on scores less than 25 REUs (e.g., see Fig G in S1 Text). The final $\Delta\Delta\Delta G$ scores were calculated as follows to reflect the effect of mutation on the open-closed equilibrium of BK channels:

$$\Delta\Delta\Delta G = (\Delta G_{fold}^{open} - \Delta G_{fold}^{closed})_{mutant} - (\Delta G_{fold}^{open} - \Delta G_{fold}^{closed})_{WT} \tag{1}$$

The same formula was used to calculate the contributions of all Rosetta energy components, as summarized in Table A in S1 Text. The Rosetta energy calculations were performed for all 16264 single mutants of BK channels represented on the structure. We also computed the effect of each WT mutation, (e.g., A316A), effectively as a baseline in comparison to the true mutants.

## Atomistic simulations and calculation of dynamical properties

The atomistic MD simulation were performed using the CHARMM36m protein [37] and CHARMM36 lipid [133] force fields using Gromacs 2019 [31,134]. The simulation was initiated from the closed state structure derived from PDB 6V3G [63]. The solvent boxes and simulation inputs were generated using CHARMM-GUI web interface [135–138]. The solvated system was energy minimized and equilibrated following the standard CHARMM-GUI Membrane Builder protocols [135,137]. The final production MD simulation was run for 100 ns with 2 fs step size. All hydrogen-containing bonds constrained with LINCS [139]. Electrostatics were treated with the particle mesh Ewald algorithm [140] with a 12 Å cutoff, and van der Waals forces were smoothly truncated from 10 to 12 Å. Nosé-Hoover thermostat [141,142] was used to maintain the simulation temperature at 303.15 K with a coupling time $\tau_T$ = 1 ps$^{-1}$. The Parrinello-Rahman semi-isotropic barostat [143] was applied in membrane lateral directions to maintain a constant pressure of 1.0 bar with compressibility of 4.5x10$^{-5}$ bar$^{-1}$ and coupling time $\tau_P$ = 5 ps$^{-1}$.

We calculated three dynamical properties from the MD trajectory with snapshots taken every 1 ns using CHARMM version 41a, including residue RMSF profile, Cα- Cα covariance matrix, and average fractions of solvent-accessible surface area (SASA) of each sidechain. The covariance matrix was calculated with CHARMM. The complete matrix corresponds to the tetramer, so the portions corresponding to the intra-monomer and directly-neighboring-monomer for residues A316 and V278, respectively. These two residues were selected because each has multiple mutations with a significant $\Delta V_{1/2}$ and because their corresponding rows of the covariance matrices were not correlated with one another. We used both the self-monomer covariance matrices and neighboring-monomer covariance matrices, and the final matrices were the average across all four monomers. We calculated RMSF of all Cα atoms using CHARMM. Finally, we calculated the percentage of solvent-exposed surface area by

calculating the SASA for each residue using CHARMM divided by the average SASA of a fully solvent-exposed residue.

## Additional structural and sequence-based features

Additional structural features were calculated from the equilibrated closed state structure using CHARMM (Table B in S1 Text). We identified residues within 5 Å of any water molecules and residues within 5 Å of any lipid molecules as water-exposed and lipid-exposed, respectively. Residues that are both water and lipid exposed are identified as those at the membrane interface (BORDER). The secondary structure of each residue (α-helix, β-sheet or coil) was assigned using the Kabsch and Sander definition [144] as implemented in CHARMM. Sequence conservation information was captured using predicted effect of single nucleotide polymorphism from SIFT [121], a genomic-level tool based on the comparison of aligned homologous sequences. This should account for the evolutionary pressure to conserve sequences like the highly-conserved selectivity filter TVGYG sequence and ignore flexible loops without strong sequence dependence across other channels. All the calculated features, its abbreviated name, a brief description, and the type of calculation are listed in Table B in S1 Text.

## Statistical model selection, training and validation

We performed a grid search of hyperparameters for a variety of regression models in the python scikit-learn library [145], including Ridge regression, SVR, KNN, RF, GP and MLP. Prior to training, we examined the correlation of all the features we calculated in the previous section and discarded features with correlation greater than 0.7, or where all mutations had the same value, were removed, namely rama_prepro, yhh_planarity, hbond_lr_bb, and ref (feature correlation matrix given in Fig I in S1 Text, feature names and descriptions given in Table A in S1 Text). We then standardized the remaining features to have a mean of zero and unit variance with the standard scalar in the scikit-learn preprocessing library. For training, we first split the 473 single mutation data set randomly into 80% training and 20% testing using the scikit-learn function train_test_split in the model_selection module. We then performed the grid search of the key model parameters listed in Table B in S1 Text using 5-fold cross validation on the training data. As implemented in the scikit-learn models (RF, Ridge, SVR, MLP), we used recursive feature elimination to train with an optimal feature set for each model [146]. The space of model hyperparameters optimized over for each model, along with the top performing parameters for each, is summarized in Table B in S1 Text. We selected the RF model, as the resulting model gave optimal results on validation data while also being the relatively insusceptible to variation in training data or choice of hyperparameters. The final RF model hyperparameters are given in Table 3. There was no significant performance boost to dropping any features from the RF model, which is expected from this type of models when the features aren't strongly correlated. The performance of the RF model on the 5-fold CV splits and the 20% independent test set is reported in Fig D in S1 Text. The enrichment factor (EF) is defined as:

$$EF = \frac{(\% \, \mathrm{of\,top\,10\,with\,} \Delta V_{1/2} > 50\mathrm{mV}) + (\% \, \mathrm{of\,bottom\,10\,with\,} \Delta V_{1/2} < -50\mathrm{mV})}{f_{50}}, \qquad (2)$$

where $f_{50} = 0.23$, the fraction of mutations greater than ± 50 mV in the full dataset.

We noticed substantial variance across different splits owing to differences in the distribution of the experimental $\Delta V_{1/2}$ between train and test sets. We expected that the true model performance might be under- or over-estimated given only a single random split. To better

**Table 3. Final RF model hyperparameters.** These parameters were selected based on initial 5-fold cross-validation analysis on the 80% training data (see Table B in S1 Text). Description of hyperparameters were adapted from SciKit-learn documentation.

| Name | Description | Value |
|------|-------------|-------|
| n_estimators | Number of estimators in model (trees) in forest | 500 |
| min_samples_split | Minimum samples to split internal node | 2 |
| max_leaf_nodes | Maximum number of leaf nodes, best first | 100 |
| max_depth | Maximum tree depth | 20 |
| max_features | Maximum features to train with | 1.0 |
| ccp_alpha | Cost-Complexity Pruning alpha | 0.01 |
| bootstrap | Bootstrap Aggregation (BAGGing) | True |
| oob_score | Out-Of-Bag Score (error estimate) | True |
| max_samples | Maximum fraction of samples to train each tree with | 1.0 |
| min_samples_leaf | Minimum number of samples in leaf | 1 |

estimate the RF model's performance if trained on the full dataset, we performed the same cross-validation on five separate 80/20 splits, and the RF model appeared relatively stable across these tests (Fig 3). The final RF hyperparameters were then used to train a production model on the complete dataset. The predictions of the production model are available as an Excel sheet in the S1 Text as well as the GitHub repository (see below). Where shown, error bars denote prediction uncertainty estimated using the bootstrap aggregation (bagging) method for random forest regression described by Wager, *et al.* [125,126].

We used python3 for all the machine learning scripts including the pandas [147], scikit-learn [145], and forestci [126] packages. All the numerical plots were generated using matplotlib [148] and all the molecular rendering using VMD [149]. The relevant scripts and data files are available at: https://github.com/enordquist/bkpred.

## Experimental methods: Oocyte expression, mutagenesis, and electrophysiology

The four novel mutations in this study (Table 4) were prepared using Pfu polymerase (Stratagene, La Jolla, CA) to overlap extension Polymerase chain reaction (PCR) from template of the mbr5 splice variant of mslo1 (Uniprot ID: Q08460). We then sequenced the PCR-amplified regions to confirm the mutations. mRNA was synthesized in vitro using T3 polymerase (Ambion, Austin, TX) from linearized cDNA. Approximately 0.05–50 or 150–250 ng/oocyte mRNA of the mutations was injected into stage IV-V oocytes from female *X. laevis*. After 3–6 days of incubation at 18˚C, the oocytes are prepared for electrophysiology recordings.

The experimental electrophysiology readings were collected from excised inside-out patches on a set-up with an Axopatch 200-B patch-clamp amplifier (Molecular Devices, Sunnyvale, CA) and ITC-18 interface with Pulse acquisition software (HEKA Electronik, Division

**Table 4. Boltzmann fit parameters of G-V curves.**

| | $V_{1/2} \pm$ SEM (mV) | $b \pm$ SEM (mV) |
|------|------|------|
| **WT** | 176.2 ± 4.4 | 29.0 ± 4.0 |
| **L235H** | 187.9 ± 3.6 | 23.9 ± 3.3 |
| **L235F** | 230.7 ± 6.3 | 27.3 ± 6.7 |
| **V236H** | 139.8 ± 4.8 | 20.0 ± 4.2 |
| **V236N** | 164.5 ± 4.9 | 19.2 ± 4.3 |

of Harvard Bioscience, Holliston, MA). We pulled the Borosilicate pipettes using a Sutter P-1000 (Sutter Instrument, Novato, CA) to obtain 0.5–1.5 MΩ resistance for inside-out patches obtained from the oocyte membrane. We obtained the current signals with low-pass-filtered at 10 KHz and digitized at 20-μs intervals. Two solutions were used in recording. The first was a pipette solution (in mM): 140 potassium methanesulphonic acid, 20 HEPES, 2 KCl, 2 MgCl2, pH 7.2. The second was the nominal 0 μM $[Ca^{2+}]_i$ solution (in mM): 140 potassium methane-sulphonic acid, 20 HEPES, 2 KCl, 5 EGTA, pH 7.1–7.2. There is about 0.5 nM free $[Ca^{2+}]_i$ in the nominal 0 $[Ca^{2+}]_i$ solution.

We measured the macroscopic tail current at either -80 mV or -120 mV to determine the relative conductance. The conductance-voltage (G-V) relationships were fitted using the Boltzmann function:

$$\frac{G(V)}{G_{max}} = \frac{1}{1 + exp\left(-\frac{ze(V - V_{1/2})}{kT}\right)} = \frac{1}{1 + exp\left(\frac{V_{1/2} - V}{b}\right)}, \qquad (3)$$

where $G/G_{max}$ represents the ratio of conductance to maximal conductance, $z$ denotes the number of equivalent charges, $e$ denotes the elementary charge, $V$ denotes the membrane potential, $V_{1/2}$ is the voltage in mV at which $G/G_{max}$ reaches 0.5, $k$ denotes Boltzmann's constant, $T$ denotes absolute temperature, and $b$ is the slope factor measured in mV. The fitted parameters are reported in Table 4. Each G-V relationship presented in the figures is the average of 3–7 patches, and the error bars indicate the standard error of means (SEM). Similar protocols have been used for the 230 previously acquired mutations that had been expressed and characterized in the Cui lab over the years.

## Supporting information

**S1 Text. Fig A. Mean RMSF (blue) and percent SASA (orange) by Residue ID (ResID).** The RMSF has the units of Å, and the mean percent SASA is a unitless percentage. These features were calculated from 100 ns MD simulation of the deactivated BK channel. **Fig B. Correlation of experimental ΔV$_{1/2}$ with the total Rosetta ΔΔΔG and LJ dispersion, the change in hydrophobicity upon mutation, and the Cα-Cα covariance with residue A316.** The Pearson correlation coefficient is R < 0.01. See method section for description of how the Rosetta ΔΔΔG scores were calculated. **Fig C. Distributions of (A) experimental ΔV$_{1/2}$ values and (B) number of existing mutations along the sequence.** Each histogram was generated using 100 bins. **(A)** depicts the ΔV$_{1/2}$ distribution after squashing between ± 200 mV (see Methods). The key functional domains are highlighted beneath **(B). Fig D. 5-fold cross-validation on initial training split.** In all panels, blue denotes a data point used to train that iteration of the model, orange denotes validation or testing. The first five panels correspond to the 5-fold cross-validation within the training set (80% of the total data). The final panel (bottom right) was trained on the full training set and validated using the rest 20% of the full data set. This split corresponds to the 80/20 training/test split 1 from Table 1. **Fig E. Feature Importance.** Importance is reported as the mean decrease in Gini impurity score. A larger decrease in this impurity score means that using the feature in a branch of a tree often leads to greater ability to distinguish large and small shifts, for example. Names of features and their source are provided in Table B in S1 Text. **Fig F. Performance of control model trained without physics-based descriptors.** In all panels, blue dots denote data points used to train that iteration of the model, and orange dots denote the independent test data. **Fig G. Correlation of the true and predicted error for the train and test sets of the five independent data splits.** Blue and orange dot represent training and test data, respectively. The dashed lines denote lines of best fit with the same color scheme. The gray lines denote x = y. **Fig H. Illustration of the**

**squashing function for pre-processing of various raw quantities.** The blue line plots the original, un-squashed function as a reference. **Fig I. Feature correlations.** Correlation of the features used in the model training. Correlation is reported as the Pearson correlation coefficient. Names and descriptions of features and their sources are provided in Table A in S1 Text. **Table A. List of features and their descriptions.** All Rosetta terms are $\Delta\Delta\Delta G$ values between the closed and open states as described in Methods. **Table B. Summary of machine learning models, hyperparameters trained in Grid Search, and training and validation correlation (R) and RMSE from 5 independent splits of data.** The set of hyperparameters which correspond to top mean score are bolded. The pairs of scores correspond to all five splits of data. **Table C. Neurological BK channel mutants, channel activity phenotypes, functional mechanisms (if known), and predicted $\Delta V_{1/2}$ values.** Abbreviations: VUS: Variant of Uncertain Significance, NE: No effect, LOF: Loss of Function, and GOF: Gain of Function. The Coordination of Rare Diseases at Sanford (CoRDS) is a standardized patient registry in a de-identified format. NP stands for No Prediction; the model doesn't predict any shift for mutations to residues absent in either PDB structure.
(PDF)

## Author Contributions

**Conceptualization:** Jianhan Chen.

**Data curation:** Erik Nordquist, Guohui Zhang, Nathan Ji, Kelli M. White, Lu Han, Zhiguang Jia, Jingyi Shi.

**Formal analysis:** Erik Nordquist, Nathan Ji, Jianhan Chen.

**Funding acquisition:** Jianmin Cui, Jianhan Chen.

**Investigation:** Shrishti Barethiya, Jianmin Cui, Jianhan Chen.

**Methodology:** Erik Nordquist, Shrishti Barethiya, Zhiguang Jia, Jianmin Cui, Jianhan Chen.

**Project administration:** Jianmin Cui, Jianhan Chen.

**Resources:** Jianmin Cui, Jianhan Chen.

**Software:** Jianhan Chen.

**Supervision:** Jingyi Shi, Jianmin Cui, Jianhan Chen.

**Validation:** Guohui Zhang, Jianhan Chen.

**Writing – original draft:** Erik Nordquist, Guohui Zhang, Jianhan Chen.

**Writing – review & editing:** Erik Nordquist, Guohui Zhang, Shrishti Barethiya, Jianmin Cui, Jianhan Chen.

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
