## [Decision Letter · Decision Letter 0]

23 Jul 2023

Dear Dr. Chen,

Thank you very much for submitting your manuscript "Incorporating physics to overcome data scarcity in  predictive modeling of protein function: a case study of BK channels" for consideration at PLOS Computational Biology.

As with all papers reviewed by the journal, your manuscript was reviewed by members of the editorial board and by several independent reviewers. In light of the reviews (below this email), we would like to invite the resubmission of a significantly-revised version that takes into account the reviewers' comments.

We cannot make any decision about publication until we have seen the revised manuscript and your response to the reviewers' comments. Your revised manuscript is also likely to be sent to reviewers for further evaluation.

Sincerely,

Alexander MacKerell

Academic Editor

PLOS Computational Biology

Nir Ben-Tal

Section Editor

PLOS Computational Biology

Reviewer's Responses to Questions

**Comments to the Authors:**

Reviewer #1: The manuscript presents a study combining physically motivated molecular features with sparse biochemical data to predict biomolecule functional transitions. This is the critical challenge that much of the molecular biophysics community faces nowadays. The authors show one interesting approach to dealing with such a challenge using ion channel gating. Using simulations and sparse but still significant experimental data, authors train a random forest model to predict gating trends. The model shows robustness in predicting global trends (hydrophobic nature of gating) and shows an impressive ability even to predict some subtle mutations on the helix of the transmembrane region.

That being said, I found some parts of the manuscript confusing and lacking in crucial details, preventing me from fully understanding some of the authors' choices with their methodology. Specifically, the following points can be improved:

The description of the dataset and all the considered features used in training must be motivated and listed early on in the results section. Having a section dedicated to i) carefully describing the dataset, ii) containing a table listing names and variables extracted from MD simulations, rosetta calculations, and iii) references and to databases and papers from where data is collected. iv) visualization of the dataset correlated with simple features to allow readers to see what the challenge is.

The paper needs to describe the choice and nature of training and test splitting choice better. These should be carefully motivated. Have authors considered various forms of scaling of heterogeneous features? This is often decisive in sparse datasets with magnitudes of features on vastly different scales.

Fig 2 I found it could have been more useful for reinforcing the main results. Could authors plot the effects of mutations on voltage on a plot and highlight train and test sets? or show a correlation with the degree of hydrophobicity.

The authors looked at Ca-Ca distances, a rather coarse measure for using in detailed atomistic effects such as changes in side chains introduced in mutations. Other atomically detailed measures could be considered, such as the number of hydrophobic clusters, hydrogen bonding networks, or salt bridges.

In the results sections, authors could place their work in the context of prior studies by reviewing the literature on using RF/ML to predict functional mutations.

Chen, A.S.Y., Westwood, N.J., Brear, P., Rogers, G.W., Mavridis, L. and Mitchell, J.B., 2016. A random forest model for predicting allosteric and functional sites on proteins. Molecular informatics, 35(3‐4), pp.125-135.

Rodrigues CH, Pires DE, Ascher DB. DynaMut: predicting the impact of mutations on protein conformation, flexibility and stability. Nucleic acids research. 2018 Jul 2;46(W1):W350-5.

Dutagaci B, Duan B, Qiu C, Kaplan CD, Feig M. Characterization of RNA polymerase II trigger loop mutations using molecular dynamics simulations and machine learning. PLOS Computational Biology. 2023 Mar 22;19(3):e1010999

Yasuda I, Endo K, Hirano Y, Yamamoto E, Yasuoka K. Relation in ligand dynamics and protein mutation using unsupervised machine learning. Biophysical Journal. 2023 Feb 10;122(3):182a.

Reviewer #2: The authors present an interesting work where they incorporated predicted physical properties from MD simulations to an existing data set of protein mutation, to build a predictive model of protein function. They used the BK channels as a case study for their proposed method.

Overall, the idea of extending existing data set with simulation data is appealing, and it is reasonable to expect it to improve the model predictive ability. Focusing this review on the modeling effort, I have a few questions:

1. I'm unfamiliar with how the mutation data is usually represented, and I don't see it anywhere explained in the paper. The author could include a brief description of this data and how it is incorporated in the ML model. The topic is interesting and adding this information may extend the readability of the paper for readers interested in combining data and simulation results.

2. The authors use the simulation data to extend the dimensionality of the input data X for training the model, without increasing the size of the data set. That in itself would not be a problem, if the original data set covered a larger fraction of possible mutations. When commenting about the performance of the model over the different splits, the authors claim that the model perform better when training and test set have similar coverage. Which raises the question, if we are covering only ~3% of your mutation space, and the coverage seems to be very important in the performance of the model. How can this model be generalized to unseen mutations.

In the same line, it would be interesting to see how the mutations are encoded in the data, and what coverage means exactly in this context.

3. Please, verify the sue of RMSE and RMSF

**Have the authors made all data and (if applicable) computational code underlying the findings in their manuscript fully available?**

Reviewer #1: Yes

Reviewer #2: Yes

PLOS authors have the option to publish the peer review history of their article (what does this mean?). If published, this will include your full peer review and any attached files.

Reviewer #1: No

Reviewer #2: No
---

## [Decision Letter · Decision Letter 1]

24 Aug 2023

Dear Dr. Chen,

We are pleased to inform you that your manuscript 'Incorporating physics to overcome data scarcity in  predictive modeling of protein function: a case study of BK channels' has been provisionally accepted for publication in PLOS Computational Biology.

Best regards,

Alexander MacKerell

Academic Editor

PLOS Computational Biology

Nir Ben-Tal

Section Editor

PLOS Computational Biology

Reviewer's Responses to Questions

**Comments to the Authors:**

Reviewer #1: Authors have thoroughly addressed all my concerns.

**Have the authors made all data and (if applicable) computational code underlying the findings in their manuscript fully available?**

Reviewer #1: Yes

PLOS authors have the option to publish the peer review history of their article (what does this mean?). If published, this will include your full peer review and any attached files.

Reviewer #1: No

---

## [Editor Report · Acceptance letter]

12 Sep 2023

PCOMPBIOL-D-23-01000R1 

Incorporating physics to overcome data scarcity in  predictive modeling of protein function: a case study of BK channels

Dear Dr Chen,

I am pleased to inform you that your manuscript has been formally accepted for publication in PLOS Computational Biology. Your manuscript is now with our production department and you will be notified of the publication date in due course.

With kind regards,

Judit Kozma
